behaviour

brood parasitism, common cuckoo, egg crypsis, egg removal, Oriental reed warbler

**Author for correspondence:**
Wei Liang
e-mail: liangwei@hainnu.edu.cn

# Common cuckoo females remove more conspicuous eggs during parasitism

Longwu Wang[1], Yuhan Zhang[1], Wei Liang[2] and Anders Pape Møller[3,4]

[1]State Forestry Administration of China Key Laboratory for Biodiversity Conservation in Mountainous Areas of Southwest Karst, School of Life Sciences, Guizhou Normal University, Guiyang 550001, People's Republic of China
[2]Ministry of Education Key Laboratory for Ecology of Tropical Islands, College of Life Sciences, Hainan Normal University, Haikou 571158, People's Republic of China
[3]Ministry of Education Key Laboratory for Biodiversity Science and Ecological Engineering, College of Life Sciences, Beijing Normal University, Beijing 100875, People's Republic of China
[4]Ecologie Systématique Evolution, Université Paris-Sud, CNRS, AgroParisTech, Université Paris-Saclay, 91405 Orsay Cedex, France

 LW, 0000-0002-7020-4397; WL, 0000-0002-0004-9707;
APM, 0000-0003-3739-4675

Avian obligate brood parasites gain an advantage by removing the eggs of the cuckoos who have already visited the nest, which can increase the chances of survival for their offspring. Conversely, to prevent their eggs from being picked up by the next parasitic cuckoo, they need to take some precautions. Egg mimicry and egg crypsis are two alternative strategies to prevent the parasitized egg from being picked up by another parasitic cuckoo. Here, we tested whether the egg crypsis hypothesis has a preventative effect when common cuckoos (*Cuculus canorus*) parasitize their Oriental reed warbler (*Acrocephalus orientalis*) hosts. We designed two experimental groups with different crypsis effects to induce common cuckoos to lay eggs and observed whether the cuckoos selectively picked up the experimental eggs with low crypsis levels in the process of parasitism. Our results supported the egg crypsis hypothesis; the observed cuckoos significantly preferred to select the more obvious white model eggs. This shows that even in an open nest, eggs that are adequately hidden can also be protected from being picked up by cuckoo females during parasitism so as to increase the survival chance of their own parasitic eggs.

## 1. Introduction

Parasites, such as cuckoos and cowbirds, lay their eggs in the nests of other birds; therefore, passing the high reproductive cost of

parental care on to their hosts [1,2]. This causes selection for the evolution of anti-parasite strategies such as nest defence and egg recognition [3–5]. An adaptive parasite has to choose the right time to quickly lay its eggs and then escape the crime scene, avoiding an attack from the host or reducing the suspicion and detection of the eggs by the host [6–8]. In addition to laying eggs in such a limited time (3–10 s) [7,8], the parasite usually removes one or two eggs from the host's nest [9–11].

So far, many hypotheses have tried to explain the behaviour of the parasite. For example, the 'mimicry improvement hypothesis' believes that female common cuckoos (*Cuculus canorus*) take away the eggs of great reed warblers (*Acrocephalus arundinaceus*) to make their parasitic eggs a closer match to the host eggs in the nest, thereby improving the rate of successful parasitism [10]. The 'free meal hypothesis' posits that the parasite can get free nutrition from the egg they picked up to supplement nutrition consumption owing to oviposition, especially urgently needed calcium [12,13]. Among the remaining six hypotheses (reviewed in [10]), one of the more interesting ones, the 'parasite competition hypothesis', has attracted our attention. It refers to the behaviour of picking up eggs as preventing the eggs of the former parasite from hatching in order to improve the chances of the successful hatching of their own eggs. This behaviour is aimed at competitors of the same species [14]. The reason is that many parasites, such as the common cuckoo, are evictor species, and parasitic cuckoo eggs will hatch 2–3 days earlier than those of the host [6,15,16]. The hatchlings will then remove all other eggs or nest-mates from the nest within a few days, leaving their own, to monopolize the care of their foster parents [16–18]. Another example is the nestlings of greater honeyguides (*Indicator indicator*), who will stab their nest-mates and even kill them with their sharp beaks to become the sole occupier of the host nest [19]. Owing to this competition process, only one parasitic nestling will remain; therefore, it is beneficial for a parasite to remove any potentially parasitic eggs in the nest. This behaviour directly eliminates the formidable enemies that this offspring may face after hatching. However, in order to prevent their eggs from being taken away by the next cuckoo of the same species, they need certain means or measures for defence.

Egg mimicry [20–22] or egg crypsis [23,24] are two alternative strategies that prevent parasitic eggs from being picked up. In egg mimicry, because all the eggs in the nest are acutely similar, the probability of the parasitic eggs being picked up is equal to that of the other eggs, and the probability is even higher when there are fewer eggs in the nest. However, in egg crypsis, because the eggs are more concealed, they are not easy for later parasites to find, thus greatly reducing the risk of being taken away. This view has been confirmed in a previous study of the little bronze-cuckoo (*Chalcites minutillus*) and its host, large-billed gerygones (*Gerygone magnirostris*) [24]. Although gerygones rarely discard artificially inserted eggs, and typically do not discard naturally parasitized cuckoo eggs, the cuckoo prefers to take away conspicuous eggs during the process of parasitism, which are five times more likely to be taken than the concealed eggs [24]. Therefore, for some species that build domed nests, under the pressure of intraspecific competition, the egg crypsis strategy is more popularly employed by the parasite to ensure that their eggs are not picked up by the late comers [24].

The common cuckoo is one of the most studied brood parasites in the world [1,2,6]. Šulc *et al.* [10] revealed that common cuckoo females are not choosy when removing an egg during parasitism and their results did not support the parasite competition hypothesis. A possible reason is that the parasitic egg is too similar to the host egg, and it is hard for the second cuckoo to identify the mimetic egg of the first cuckoo in a very limited time [10]. Contrary findings suggested that the parasite can selectively remove non-mimetic eggs, such as little bronze cuckoos that can recognize and pick up eggs with high brightness from the nest [24], and greater honeyguides prefer to puncture eggs that are larger and are different from those of the host [25]. According to the principle of the egg crypsis hypothesis mentioned above, conspicuous eggs are easier to pick up, while the more concealed eggs with low brightness are relatively safe, which has been confirmed in domed nests [23,24]. Some cuckoo females encounter other cuckoo eggs in nests with multiple occurrences of parasitism (29%, 11 out of 38). In fact, even in open nests, parasitic cuckoo eggs do not exactly match the appearance of the host egg [26–29]; the brightness of cuckoo eggs is usually higher than that of host eggs by spectral analysis [27,30,31], which means that cuckoo eggs are more prominent in the nest. In this case, is the common cuckoo inclined to pick up the more conspicuous eggs in the process of laying eggs?

The purpose of this study was to further test whether the egg crypsis hypothesis is equally applicable in the open nest parasitic system. We took a proof-of-concept approach to testing the idea that, in open nests, the more cryptic the egg, the less likely it is to be removed by a cuckoo, and the cuckoo will selectively pick up the more conspicuous eggs. In order to make the eggs more concealed or more conspicuous in the nest, we set up two groups of test nests with different crypsis effects, using black

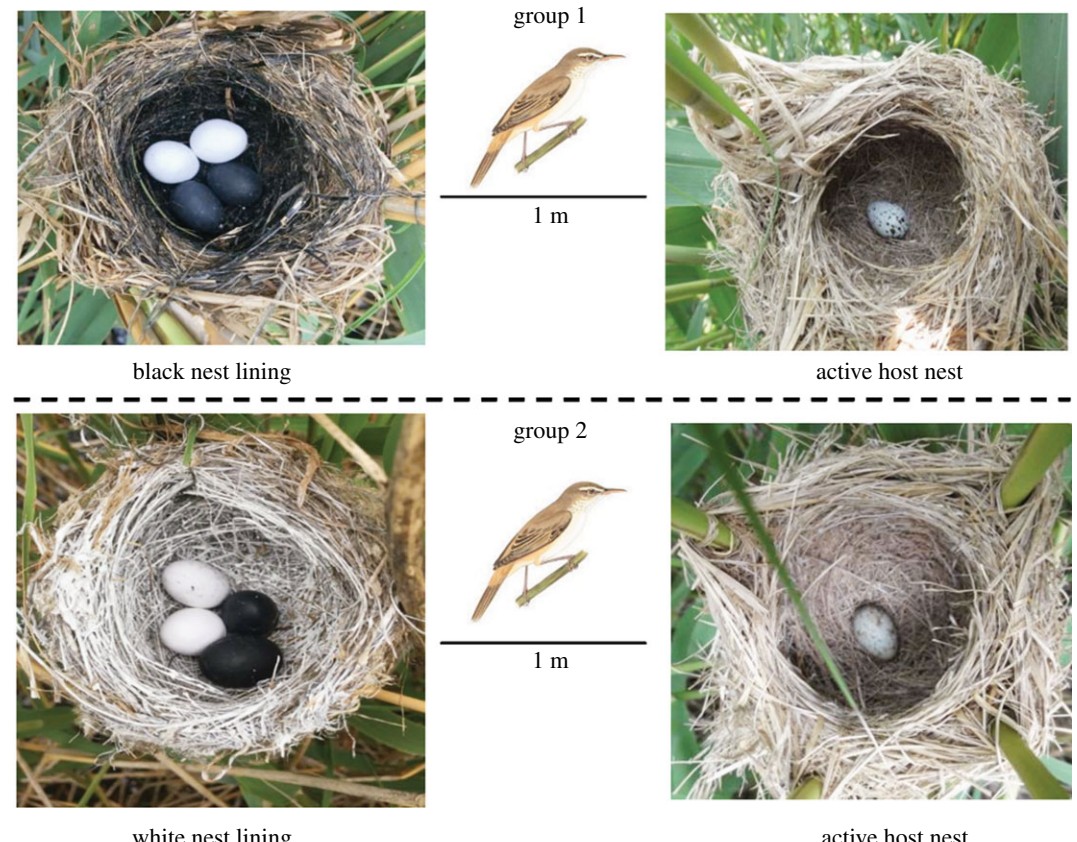

**Figure 1.** Example of the experimental nests used for eliciting cuckoo parasitism in this study. Groups 1 and 2 show black or white nest lining with model eggs, respectively.

and white model eggs and the lining of the nest to mimic the two effects of conspicuous and concealed eggs (figure 1). We used these test nests to attract common cuckoos to parasitize them and observed whether the common cuckoo could selectively pick up the eggs according to their degree of crypsis. We assumed that the white and black model eggs would show a very different crypsis effect under the background of black or white nests, and the more conspicuous the eggs, the more likely they would be picked up by the cuckoo female. To the best of our knowledge, this is the first experiment to test the egg crypsis hypothesis by setting artificial nests to observe how cuckoos pick up eggs.

# 2. Material and methods

## 2.1. Study site and study species

We performed this study in Zhalong National Nature Reserve (46°48′–47°31′ N, 123°51′–124°37′ E) in Heilongjiang, Northeast China. Field experiments were carried out during the breeding season (June to August) in 2017 and 2019. Here, we systematically searched for Oriental reed warbler nests in reed habitats in our study area; the common cuckoo mainly parasitizes their host, Oriental reed warblers, who suffer a high parasitism rate of *ca* 50% (for more details, see [28,29]).

## 2.2. Field experiments

When we found a new warbler nest, which were checked daily during their egg-laying stage, we quantified the first egg-laying date, clutch size and parasitism status. Once warblers started to lay their first egg in a fully built nest, two groups of experimental nests were set up to elicit cuckoo females to approach and then parasitize them. All experimental nests came from old warbler nests that we collected in the field in the previous year after the breeding season was complete.

In group 1, we tied a prepared test nest 1 m away from the active host nest. The inner part of the nest was painted black (figure 1; group 1), dried for more than a week, and then used for the experiment

**Table 1.** Colour difference ($\Delta E$) between the white and black model eggs in the artificial painted nest lining ($N$ refers to nest sample size, values show mean ± s.d.).

| nest colour | $\Delta E$ white model egg | black model egg | $N$ | $F$ | d.f. | $p$ |
|---|---|---|---|---|---|---|
| black nest lining | 198.64 ± 35.84 | 23.16 ± 12.05 | 15 | 323.119 | 1, 28 | <0.001 |
| white nest lining | 70.00 ± 16.11 | 79.02 ± 20.74 | 15 | 0.233 | 1, 28 | 0.633 |

when there was no odour. Two black and two white model eggs made of synthetic clay (mean ± s.d., length: 30.55 ± 0.50 mm, width: 21.78 ± 0.33 mm, $n = 15$) were placed in the nest. The white eggs were very conspicuous against the black background, while the black ones were concealed, which was convenient for evaluating the cuckoos' choice. In this group of experiments, we assumed that cuckoos would pick up more white eggs.

In group 2, to make the black eggs conspicuous and the white ones concealed, we painted the nest bottom white, and again added black and white model eggs. All other designs were similar to those in group 1 (figure 1; group 2). We assumed that the results of this group should be contrary to those of group 1, and that the cuckoos would pick up more black eggs.

We video recorded some experimental nests, which helped us to review whether the white or black colour model eggs were removed by the cuckoo female; if we did not film the nest which was parasitized by the cuckoo, we checked the number of model eggs and recorded the egg-removal results. To avoid any warblers removing the experimental model eggs, we purposely enlarged the size of the model eggs to one which warblers could not remove, but which cuckoos could remove easily. Therefore, even without video recording, we could correctly judge the egg removal result. We ended the attraction experiment when the warblers' clutch size was complete.

To quantify the degree of crypsis of the eggs in the black and white nests, we measured the differences in colour between the model eggs and the nest background. A Canon EOS 20D camera (Canon Inc., Tokyo, Japan) was used to take photos of the experimental nests. Finally, 15 photos of the parasitized black or white nests were selected, which resulted in 30 black and white model eggs in total. CIE$L^*a^*b^*$ (International Commission on Illumination) colour spaces [32,33] in Adobe PHOTOSHOP CS6 software were used to measure the colour of the nest linings and model eggs. The $L^*$ value represented brightness, while $a^*$ and $b^*$ were chroma, and their values represented the colour ranges from red to green and yellow to blue, respectively. For egg measurement, we used the whole picture to obtain all parameters of the model egg. For background colour, we measured four squares of the same size around the egg and calculated the average value [34]. After obtaining the values of $L^*$, $a^*$ and $b^*$, the overall colour difference ($\Delta E$) between the model egg and the nest lining was calculated according to the formula for colour difference, which can be used to indicate the crypsis degree of the eggs. The calculation formula is as follows:

$$\Delta E = \sqrt{(L_m^* - L_n^*)^2 + (a_m^* - a_n^*)^2 + (b_m^* - b_n^*)^2}.$$

In the formula, $m$ stands for model egg, $n$ stands for nest lining, and the smaller $\Delta E$ is, the more concealed the model egg is. The calculation results are listed in table 1.

## 2.3. Statistical analysis

Fisher's exact tests were used to estimate the frequency of egg removal during the cuckoo's egg-laying. Fisher's exact tests were used if the effective sample size was less than five. One-way ANOVA was used to test for a colour difference ($\Delta E$) in the black/white nest lining. Differences were considered to be significant at the 0.05 level. The statistical analysis was conducted using IBM SPSS Version 25.0 (IBM Corp., Armonk, NY, USA). Values are presented as mean ± standard deviation (s.d.) unless stated otherwise.

## 3. Results

For group 1, the treatment with the black nest lining, 38 nests were set up in total, and 16 nests were successfully parasitized; among them, cuckoos picked up white eggs in nine nests and black eggs in

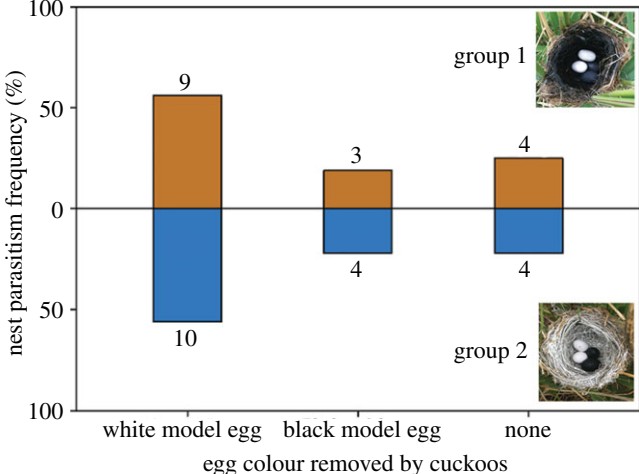

**Figure 2.** Frequency distribution of cuckoos choosing experimental nests in relation to the colour of the model eggs in groups 1 and 2 when parasitism occurred. Numbers above the bars refer to sample size.

three nests, but did not pick up any eggs in four nests (figure 2, group 1). The parasitism rate was 42.1% (16 out of 38). Cuckoo females preferred to pick up white eggs (Fisher's exact test, $p = 0.039$). For group 2, the treatment with the white nest lining, 34 nests were set up in total, and 18 nests were successfully parasitized; among them, the cuckoos picked up white eggs in 10 nests and black eggs in four nests, but did not pick up any eggs in four nests (figure 2, group 2). The parasitism rate was 52.9% (18 out of 34). Although there was no significant difference in the preference of picking up black and white eggs (Fisher's exact test, $p = 0.057$), cuckoo females still preferred to pick up white eggs.

For the colour difference ($\Delta E$) in the black nest lining (table 1), the mean $\Delta E$ of the white model versus black model eggs was 198 : 23; the white eggs were more conspicuous than the black ones (ANOVA, $F_{1,28} = 323.119$, $p < 0.001$). For $\Delta E$ in the white nest lining (table 1), the mean $\Delta E$ of the white model versus black model eggs was 70 : 79; both colours of model eggs in the nest did not differ in crypsis (ANOVA, $F_{1,28} = 0.233$, $p = 0.633$), which was against our prediction that the black model eggs would be more conspicuous than the white model eggs against the white nest lining.

In addition, all cases of parasitism occurred during the egg-laying stage, and no parasitism occurred after fledging. Neither of the two groups had cases of parasitism in active host nests, nor did the experimental treatment cause warblers to abandon their nests. Five video recordings showed both no removal and cuckoo egg removal of the white and black model eggs in group 1 (electronic supplementary material, videos S1–S3) and no removal and egg removal of the white model eggs in group 2 (electronic supplementary material, videos S4 and S5).

# 4. Discussion

The main finding of this study was that common cuckoo females preferred to pick up the white model eggs with high brightness, regardless of whether those eggs were in the black or white painted nests. This shows that eggs with low crypsis level in the nest are more easily picked up by common cuckoos. Although in group 2 we expected that the white model eggs would be more concealed, while the black ones would be more conspicuous and easier to be picked up by cuckoos, the results were not consistent with our prediction. The reason is that in the nest with the white lining, there was no difference in the degree of crypsis between the two types of model eggs, which eliminates concealment of white model eggs, while the black model was not completely conspicuous. In any case, from group 1, our results supported the egg crypsis hypothesis that the more conspicuous the egg, the more likely it is to be removed by common cuckoos.

The non-mimetic eggs in the nest could easily be picked up by later parasitic cuckoos. This behaviour is beneficial for the parasites, especially for the species that eject eggs and nestlings from nests after hatching. However, for the initial parasite, this behaviour is nothing but a disaster. Therefore, in order to avoid intraspecific competition, parasitic individuals usually occupy a certain parasitic territory, where the cuckoo female can select all available host nests within an area. This strategy could reduce the reproductive waste caused by multiple parasitism [35,36]. Conversely, common cuckoos can

produce more concealed eggs, so that later laying cuckoo females have difficulty finding their eggs in a limited time. For example, Australian bronze cuckoos (*Chalcites* spp.) lay olive-brown cuckoo eggs which are very different from the white spotted host eggs, but are still successfully accepted by the host [23]. Experiments also show that eggs with low brightness are rarely picked up by females of these parasite species [24]. Therefore, in group 1, the black eggs were more concealed because they were almost the same colour as the background nest lining. By contrast, the white eggs were extremely conspicuous against the black background, and most were removed by the cuckoos, which is consistent with the egg crypsis hypothesis. This result was inconsistent with the random selection theory proposed by Šulc *et al*. [10]. A possible reason is that, as they explained, cuckoo eggs and host eggs are very similar, which makes it difficult for cuckoos to recognize and pick up parasitic eggs in a quite limited time. The model eggs used in this study were significantly different from the host eggs, which made them easily recognized and selected by cuckoo females. The more conspicuous, the easier to be selected.

In experimental group 2, in theory, with the white background, cuckoo females should have picked up more black eggs than white ones; however, the result was just the opposite. Some possible explanations are as follows: (i) the white model eggs on the white nest lining were not as concealed as we imagined; their degree of crypsis was similar to that of the black ones (there was no significant difference in $\Delta E$). Theoretically, the probability of the black and white eggs being removed should be the same. However, there was still a marginally significant difference between them, showing that the white eggs were preferred by the cuckoo females. Birds can detect ultraviolet (UV) light besides visible light by using a fourth cone cell type in the retina [37–40]. However, in this study, all the model eggs were made of synthetic clay, which have no UV reflectance spectra (300–400 nm) in egg appearance [41,42]; in such a case, the cuckoo cannot selectively remove the model egg based on a UV cue. The colour difference ($\Delta E$) in this study was enough to reveal the degree of crypsis for black and white model eggs in the host nests; and (ii) most eggs in nature are white, blue, or brown. There are few reports of pure black eggs [43]. The eggs of Oriental reed warblers and common cuckoos are both white with brown spots [28,29]. It is possible that common cuckoos prefer similar eggs, so they choose white ones; this reason could also be compatible with the results for group 1. We know that a few warblers have albinism when they lay their last egg, i.e. no spots (reviewed in [44]).

In all experimental cases, neither of the two groups had cases of parasitism in active host nests. This is because all the decoy nests were fixed higher than the active host nests; when the cuckoo females glide toward these nests, they first found the higher decoy nests and quickly parasitized them rather than the active host nests. Another possibility was that the host would strongly attack or mob the cuckoo, who had no more time (normally 3–10 s) to recognize the real host nests and make the right choice, given that the cuckoos could be so readily fooled by the decoy nests.

In conclusion, our results supported the egg crypsis hypothesis, and proved that the more conspicuous eggs were easily picked up in the process of cuckoo parasitism, while the more concealed eggs had a higher probability of survival. This study allowed us to further understand why the brood parasite lays more concealed eggs. The advantage of this behaviour system is that it can greatly improve the probability of having its egg accepted by the host while not being picked up by another competitor. In this study, the egg crypsis hypothesis was verified by our experiments of observing the egg-picking behaviour of cuckoos during parasitism. However, our experiment could not fully confirm the crypsis hypothesis for all parasitic systems. Moreover, whether common cuckoos have a preference for differently sized or differently coloured eggs requires further study.

Ethics. The experiments complied with the current laws of China, where they were performed. Fieldwork was carried out with permission (No. ZL-GZNU-2019-06) from the Zhalong National Nature Reserve, Heilongjiang, China. Experimental procedures were in agreement with the Animal Research Ethics Committee of Hainan Provincial Education Centre for Ecology and Environment, Hainan Normal University (no. HNECEE-2012-003).

Data accessibility. We submitted five videos, three Excel data and two figures refer to our study as electronic supplementary material.

Authors' contributions. W.L. and A.P.M. discussed and designed the study and helped improve the manuscript; L.W. and Y.Z. carried out the field experiments; L.W. performed the statistical analyses and wrote the draft manuscript. All authors approved the final submission.

Competing interests. We declare that we have no competing interests.

Funding. This work was supported by the National Natural Science Foundation of China (grant nos 31660617 and 31960105 to L.W., grant nos 31772453 and 31970427 to W.L.). L.W. was funded by a PhD grant from Guizhou

Normal University (grant no. 0516009) and new seedling plans of Guizhou Normal University (grant nos 2019, 20175726-51) and the Joint Fund of the National Natural Science Foundation of China and the Karst Science Research Center of Guizhou province (grant no. U1812401).

Acknowledgements. We are grateful to Wenfeng Wang and Jianhua Ma from Zhalong National Nature Reserve for their help and cooperation.

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
