## [Reviewer comments · Royal Society Open Science]

Review History

RSOS-200226.R0 (Original submission)

Review form: Reviewer 1

Is the manuscript scientifically sound in its present form?

Yes

Are the interpretations and conclusions justified by the results?

No

Is the language acceptable?

No

Do you have any ethical concerns with this paper?

No

Have you any concerns about statistical analyses in this paper?

No

Recommendation?

Major revision is needed (please make suggestions in comments)

Comments to the Author(s)

This study continues a long tradition of using model-egg experiments to gain insight into the behavioural ecology of avian brood parasites and their hosts. In this case, the aim is to assess whether common cuckoos are more likely to remove the most conspicuous egg from a clutch during their visit to host nests. The logic behind this test is that, when multiple cuckoos target the same host nest, selection should favour (i) cuckoos that can recognise and preferentially remove previously-laid cuckoos eggs over host eggs, and in turn (ii) cuckoo egg traits that reduce the risk of removal by other cuckoos.

The experimental design assesses egg removal by cuckoos of black vs white model eggs in both black-painted nests (where white eggs should be most visible) and white-painted nests (where we might expect black eggs to be most visible). The design turns out to be less reciprocal in practice, however, than intended, because white and black eggs are similarly contrasting in white nests. This ultimately complicates somewhat the interpretation of the results, because cuckoos most often removed white eggs in both types of experimental nest. Nevertheless, I think there is interesting data here which adds to our understanding of cuckoo behaviour.

Some points to consider if revising the manuscript:

Two possible explanations for the results are given in the Discussion: cuckoos may just have a visual template for white (or whitish) eggs (irrespective of background contrast), or cuckoos tend to choose the more conspicuous eggs. The latter is emphasized as the likely correct explanation (i.e. in support of the idea that conspicuous eggs are more vulnerable to removal by cuckoos), but the two seem equally plausible to me from the available data. I think both explanations reveal something about cuckoo egg removal behaviour worth discussion.

I also got a bit lost on the logic of testing this egg crypsis hypothesis in a system where multiple parasitism (i.e. two females cuckoos targeting the same host nest) is infrequent. Perhaps it would be clearer if, at the end of the Introduction, it is explicitly stated that your experiments take a proof-of-concept approach to testing the idea that, in open nests, the more cryptic the egg the less likely it is to be removed by a cuckoo. Perhaps you could also cite examples from other parts of the large range of common cuckoos where multiple parasitism of hosts does occur? Or the question could be framed as a general one about the sensory constraints that brood parasites face, given that they must keep visits to the host nest very brief.

There is a lot of imprecise language throughout the manuscript, which I think would make it difficult to follow for any readers not already familiar with these systems and hypotheses. In most cases these can be remedied with some rephrasing and careful consideration of words. That is, the ideas in the manuscript are well-structured, so it is just revision of the language itself that is needed.

Unfortunately I couldn't view the videos as I couldn't figure out how to open rar files on my mac. A disappointment because I'm sure they are fascinating!

Other comments:

1: It might be worth revisiting the title. My view is that "prefer" is not the right word, because it implies that female cuckoos see/sense all eggs and make a choice, which is not the same as selecting cryptic eggs less often because they are harder to see.

109 It would be worth pointing out somewhere in the Introduction that predation is a major selective force on egg appearance. Cuckoos are really just a special category of egg-predator (in that they predate a single egg from a clutch only).

135: "very different" is a bit subjective; Honeyguide eggs in this system differ in size and shape from host eggs, but all are white.

140: Here there needs to be some info on the rates of multiple parasitism in common cuckoos. How often do they encounter other cuckoo eggs in the nets?

186: Were the eggs non-toxic, given that cuckoos eat them? Were the model eggs made of plaster?

186: What size are the eggs? Were they modelled in size and shape on common cuckoo eggs?

200: Were the photos taken in situ in the field once nests were set-up? Or under some standardised lab condition? I could imagine the ambient light etc might affect the relative conspicuousness of eggs in a nest.

205: This seems like a reasonable objective measure of colour difference, and probably captures well enough the relative conspicuousness of each egg type. However, it is worth noting here that birds perceive colour quite differently to us, and that previous work – including on common cuckoos – has modelled how egg colouration is perceived by birds e.g. Langmore et al (2009) *Anim Behav*; Stoddard & Stevens (2011) *Evol*.

198: Were similar photos/measurements made for natural eggs and nests? The photos in Fig 1 suggest that the painted-black nests are not so different to natural nests, because in the latter the relatively deep nest cup creates a darkened shadow background.

257: "Neither of the two groups had cases of parasitism in active host nests.." This is quite remarkable! Who would have guessed that cuckoos could be so readily fooled by decoy nests?! Given that some hosts of brood parasites across the world use various forms of decoy nests as anti-parasite defences, it would be interesting to make a note of this in your Discussion.

262: What about predation by other animals? Were any of the nests in either treatment predated (all eggs removed), or the active nests near to them predated?

312: Given that white eggs were removed most often in both nest types, and that white and black eggs were similarly conspicuous in white nests, I think it is worth leaving the interpretation of results a little more open. To rule out the possibility that cuckoos just tend to remove white eggs (irrespective of contrast/conspicuousness) you would need a 3rd treatment in which there were white eggs + bright eggs of another colour (say blue) and show that there was no preference in removal.

318: The closing paragraph about the relative benefits of crypsis vs host-mimicry needs a bit more clarity. In systems where host defences are the main selective pressure on cuckoo egg appearance, then egg mimicry is likely to evolve. In systems where other parasites are the main selective pressure, then crypsis is a better strategy than mimicry, because it increases the odds of a cuckoo egg's survival above chance levels.

The situation in open nesting hosts is an interesting one to consider, because natural selection is presumably already pushing host eggs to be relatively cryptic e.g. speckled rather than plain white, as a way to hide them from predators. Cuckoo eggs are subject to this selective pressure

from predation too (as well as needing to be mimetic if hosts are rejectors). In short, selection on cuckoo egg morphology is coming from many directions: predators, hosts and other cuckoos.

Review form: Reviewer 2

Is the manuscript scientifically sound in its present form?

Yes

Are the interpretations and conclusions justified by the results?

Yes

Is the language acceptable?

Yes

Do you have any ethical concerns with this paper?

No

Have you any concerns about statistical analyses in this paper?

No

Recommendation?

Major revision is needed (please make suggestions in comments)

Comments to the Author(s)

The author established experiments to assess whether the egg cryptic hypothesis is applicable in the open nests parasitic system, and especially to test whether the female common cuckoos selectively remove more conspicuous eggs than cryptic eggs. In the first experiments, female cuckoos were more likely to pick up more conspicuous white eggs on the black nest lining. In the second experiment, there was no difference in the degree of crypsis between white and black model eggs on the white nest lining. Nevertheless, cuckoos were still more likely to remove white model eggs. The author concluded that egg cryptic hypothesis is applicable in the type of open nests, especially to avoid the rejection by the second female.

The experiment itself is very interesting and I think that most parts of this manuscript were quite well-written. However, I have some concerns - the interpretation/discussion of the results.

In both experiments, white model eggs were more likely to be rejected, and this may be simply caused of the preference of the female cuckoos; the warbler eggs are white with brown spots, so female cuckoos may just prefer removing the white model eggs. The author described this point for only the second experiment in the discussion, but I think that it can be explained for both experiments. Discussion need to discuss each experiment, and also tie it all together and interpret all the results.

The author may need to do an additional experiment; use different egg colour instead of white.

However, I fully understand that how much effort/time you do need for the further study/experiment, so I encourage the author to address the limitation of the experiments and to discuss further rather than to conclude that the results support the egg crypsis hypothesis.

The other limitation of this study may be about the lack of recognition ability for the cryptic eggs in common cuckoos-hosts system. Most host species of common cuckoo build open nests.

Therefore, the common cuckoo may not have evolved the rejection system against cryptic cuckoo eggs.

Minor comments:

Line 313: orient reed warbler -> oriental reed warbler

Lines 274-276: this only can be explained with the first experiment.
Line 323: behaviour -> system

Decision letter (RSOS-200226.R0)

14-Apr-2020

Dear Dr Wang:

Manuscript ID RSOS-200226 entitled "Dark eggs matter: common cuckoo females prefer to remove conspicuous eggs during parasitism" which you submitted to Royal Society Open Science, has been reviewed. The comments from reviewers are included at the bottom of this letter.

In view of the criticisms of the reviewers, the manuscript has been rejected in its current form. However, a new manuscript may be submitted which takes into consideration these comments.

Please note that resubmitting your manuscript does not guarantee eventual acceptance, and that your resubmission will be subject to peer review before a decision is made.

Your resubmitted manuscript should be submitted by 12-Oct-2020. If you are unable to submit by this date please contact the Editorial Office.

on behalf of Dr Sean Rands (Associate Editor) and Kevin Padian (Subject Editor)
openscience@royalsociety.org

Editor comments:

Thank you for your submission. As you see, several major concerns were raised, and in addition, the AE and one reviewer note that the English needs to be professionalized before we can consider publication. (I agree: it is understandable but not yet publishable.) To give you time to do this, we provide the "reject/resubmit" decision, and we wish you best success in revising. Associate Editor Comments to Author (Dr Sean Rands):

Firstly, please accept my apologies for the delay in getting a decision to you, due to the current pandemic. I hope you are well and coping with things.

Two reviewers have commented on your manuscript, and their comments should be attached. I have also read your manuscript independently of their comments, and give some additional suggestions below. Based on these three readings, I think the manuscript has potential, but you need to make some major changes to the text before this would be suitable for publication. Critically, I also think you need to make it clear that you know cuckoos were taking the eggs (see my comments to line 175), as you don't mention how you did this in the text. Please address all these comments if you decide to revise the manuscript (although I don't think that you need to conduct an additional experiment, as is suggested by reviewer #2).

44: This first sentence needs to mention who it is that is gaining an advantage by removing eggs.

49: remove 'mainly'

87: need a reference somewhere (here or near 77) that cites where a reader can learn about these other six hypotheses.

92: explain clearly what 'exclusive' means (it's vague here)

100: The 'therefore' doesn't follow from the previous, which is not giving an argument in support of it being beneficial to remove eggs. You need to add a sentence or two near here explaining why it is beneficial first.

104: Replace 'female' with 'cuckoo' - it may not be obvious to a casual reader that it is the female that does this behaviour (although of course it couldn't be the male!)

110 and 113: use 'egg mimicry' and 'egg crypsis' rather than 'former' or 'latter'

109: remove 'From a statistical point of view': statistics really refer to past events, whilst probability is more appropriate when you are talking about future events.

125: "... most studied "brood" parasites..."

175: What evidence do you have that egg removal was being done by *cuckoos* here? I would guess that you are looking for the presence of cuckoo eggs as confirmation, but you don't explicitly say this here.

198: Did you just consider human-visible colour space here? I would assume that cuckoos have similar (tetrachromatic?) vision to other birds, and so human-invisible ultraviolet colour space will also be important? You should at least acknowledge this issue in the discussion.

238: remove 'significantly' (the p value says this, so you don't need to).

224: You don't present any chi square tests in the results, so your analysis details should be adjusted.

244: The non-significant difference is for a preference, not a probability.

249: remove 'highly significantly' -the p value says this for you. Please give the two degrees of freedom for the F value.

252: remove 'significantly' and give degrees of freedom.

282: 'certain parasitic field' is very vague - please clarify.

Table 1: need both degrees of freedom for each colour types. Please state in the legend what the reported colour differences are (I'd assume mean \pm SEM?)

Supplementary Material: are you going to include your raw data as well, so that it can be reused?

Associate Editor: 2

Comments to the Author:

(There are no comments.)

Reviewers' Comments to Author:

Reviewer: 1

Comments to the Author(s)

This study continues a long tradition of using model-egg experiments to gain insight into the behavioural ecology of avian brood parasites and their hosts. In this case, the aim is to assess whether common cuckoos are more likely to remove the most conspicuous egg from a clutch

during their visit to host nests. The logic behind this test is that, when multiple cuckoos target the same host nest, selection should favour (i) cuckoos that can recognise and preferentially remove previously-laid cuckoos eggs over host eggs, and in turn (ii) cuckoo egg traits that reduce the risk of removal by other cuckoos.

The experimental design assesses egg removal by cuckoos of black vs white model eggs in both black-painted nests (where white eggs should be most visible) and white-painted nests (where we might expect black eggs to be most visible). The design turns out to be less reciprocal in practice, however, than intended, because white and black eggs are similarly contrasting in white nests. This ultimately complicates somewhat the interpretation of the results, because cuckoos most often removed white eggs in both types of experimental nest. Nevertheless, I think there is interesting data here which adds to our understanding of cuckoo behaviour.

Some points to consider if revising the manuscript:

Two possible explanations for the results are given in the Discussion: cuckoos may just have a visual template for white (or whitish) eggs (irrespective of background contrast), or cuckoos tend to choose the more conspicuous eggs. The latter is emphasized as the likely correct explanation (i.e. in support of the idea that conspicuous eggs are more vulnerable to removal by cuckoos), but the two seem equally plausible to me from the available data. I think both explanations reveal something about cuckoo egg removal behaviour worth discussion.

I also got a bit lost on the logic of testing this egg crypsis hypothesis in a system where multiple parasitism (i.e. two females cuckoos targeting the same host nest) is infrequent. Perhaps it would be clearer if, at the end of the Introduction, it is explicitly stated that your experiments take a proof-of-concept approach to testing the idea that, in open nests, the more cryptic the egg the less likely it is to be removed by a cuckoo. Perhaps you could also cite examples from other parts of the large range of common cuckoos where multiple parasitism of hosts does occur? Or the question could be framed as a general one about the sensory constraints that brood parasites face, given that they must keep visits to the host nest very brief.

There is a lot of imprecise language throughout the manuscript, which I think would make it difficult to follow for any readers not already familiar with these systems and hypotheses. In most cases these can be remedied with some rephrasing and careful consideration of words. That is, the ideas in the manuscript are well-structured, so it is just revision of the language itself that is needed.

Unfortunately I couldn't view the videos as I couldn't figure out how to open rar files on my mac. A disappointment because I'm sure they are fascinating!

Other comments:

1: It might be worth revisiting the title. My view is that "prefer" is not the right word, because it implies that female cuckoos see/sense all eggs and make a choice, which is not the same as selecting cryptic eggs less often because they are harder to see.

109 It would be worth pointing out somewhere in the Introduction that predation is a major selective force on egg appearance. Cuckoos are really just a special category of egg-predator (in that they predate a single egg from a clutch only).

135: "very different" is a bit subjective; Honeyguide eggs in this system differ in size and shape from host eggs, but all are white.

140: Here there needs to be some info on the rates of multiple parasitism in common cuckoos. How often do they encounter other cuckoo eggs in the nests?

186: Were the eggs non-toxic, given that cuckoos eat them? Were the model eggs made of plaster?

186: What size are the eggs? Were they modelled in size and shape on common cuckoo eggs?

200: Were the photos taken in situ in the field once nests were set-up? Or under some standardised lab condition? I could imagine the ambient light etc might affect the relative conspicuousness of eggs in a nest.

205: This seems like a reasonable objective measure of colour difference, and probably captures well enough the relative conspicuousness of each egg type. However, it is worth noting here that birds perceive colour quite differently to us, and that previous work – including on common cuckoos – has modelled how egg colouration is perceived by birds e.g. Langmore et al (2009) *Anim Behav*; Stoddard & Stevens (2011) *Evol.*

198: Were similar photos/measurements made for natural eggs and nests? The photos in Fig 1 suggest that the painted-black nests are not so different to natural nests, because in the latter the relatively deep nest cup creates a darkened shadow background.

257: “Neither of the two groups had cases of parasitism in active host nests..” This is quite remarkable! Who would have guessed that cuckoos could be so readily fooled by decoy nests?! Given that some hosts of brood parasites across the world use various forms of decoy nests as anti-parasite defences, it would be interesting to make a note of this in your Discussion.

262: What about predation by other animals? Were any of the nests in either treatment predated (all eggs removed), or the active nests near to them predated?

312: Given that white eggs were removed most often in both nest types, and that white and black eggs were similarly conspicuous in white nests, I think it is worth leaving the interpretation of results a little more open. To rule out the possibility that cuckoos just tend to remove white eggs (irrespective of contrast/conspicuousness) you would need a 3rd treatment in which there were white eggs + bright eggs of another colour (say blue) and show that there was no preference in removal.

318: The closing paragraph about the relative benefits of crypsis vs host-mimicry needs a bit more clarity. In systems where host defences are the main selective pressure on cuckoo egg appearance, then egg mimicry is likely to evolve. In systems where other parasites are the main selective pressure, then crypsis is a better strategy than mimicry, because it increases the odds of a cuckoo egg's survival above chance levels.

The situation in open nesting hosts is an interesting one to consider, because natural selection is presumably already pushing host eggs to be relatively cryptic e.g. speckled rather than plain white, as a way to hide them from predators. Cuckoo eggs are subject to this selective pressure from predation too (as well as needing to be mimetic if hosts are rejectors). In short, selection on cuckoo egg morphology is coming from many directions: predators, hosts and other cuckoos.

Reviewer: 2

Comments to the Author(s)

The author established experiments to assess whether the egg cryptic hypothesis is applicable in the open nests parasitic system, and especially to test whether the female common cuckoos

selectively remove more conspicuous eggs than cryptic eggs. In the first experiments, female cuckoos were more likely to pick up more conspicuous white eggs on the black nest lining. In the second experiment, there was no difference in the degree of crypsis between white and black model eggs on the white nest lining. Nevertheless, cuckoos were still more likely to remove white model eggs. The author concluded that egg cryptic hypothesis is applicable in the type of open nests, especially to avoid the rejection by the second female.

The experiment itself is very interesting and I think that most parts of this manuscript were quite well-written. However, I have some concerns - the interpretation/discussion of the results.

In both experiments, white model eggs were more likely to be rejected, and this may be simply caused of the preference of the female cuckoos; the warbler eggs are white with brown spots, so female cuckoos may just prefer removing the white model eggs. The author described this point for only the second experiment in the discussion, but I think that it can be explained for both experiments. Discussion need to discuss each experiment, and also tie it all together and interpret all the results.

The author may need to do an additional experiment; use different egg colour instead of white.

However, I fully understand that how much effort/time you do need for the further study/experiment, so I encourage the author to address the limitation of the experiments and to discuss further rather than to conclude that the results support the egg crypsis hypothesis.

The other limitation of this study may be about the lack of recognition ability for the cryptic eggs in common cuckoos-hosts system. Most host species of common cuckoo build open nests.

Therefore, the common cuckoo may not have evolved the rejection system against cryptic cuckoo eggs.

Minor comments:

Line 313: orient reed warbler -> oriental reed warbler

Lines 274-276: this only can be explained with the first experiment.

Line 323: behaviour -> system

Author's Response to Decision Letter for (RSOS-200226.R0)

See Appendix A.

RSOS-201264.R0

Review form: Reviewer 1

Is the manuscript scientifically sound in its present form?

Yes

Are the interpretations and conclusions justified by the results?

Yes

Is the language acceptable?

Yes

Do you have any ethical concerns with this paper?

No

Have you any concerns about statistical analyses in this paper?

No

Recommendation?

Accept as is

Comments to the Author(s)

The authors have addressed most of the comments from the previous review and the manuscript is improved. I think it makes a good contribution to our understanding of cuckoo behaviour when removing eggs at host nests.

Decision letter (RSOS-201264.R0)

Dear Dr Wang

On behalf of the Editors, we are pleased to inform you that your Manuscript RSOS-201264 "Dark eggs matter: common cuckoo females remove more conspicuous eggs during parasitism" has been accepted for publication in Royal Society Open Science subject to minor revision in accordance with the referees' reports. Please find the referees' comments along with any feedback from the Editors below my signature.

Please submit your revised manuscript and required files (see below) no later than 7 days from today's (ie 23-Nov-2020) date. Note: the ScholarOne system will 'lock' if submission of the revision is attempted 7 or more days after the deadline. If you do not think you will be able to meet this deadline please contact the editorial office immediately.

Kind regards,

Royal Society Open Science Editorial Office
Royal Society Open Science
openscience@royalsociety.org

on behalf of Dr Sean Rands (Associate Editor) and Kevin Padian (Subject Editor)
 openscience@royalsociety.org

Editor comments:

Thanks for your resubmission; it will be acceptable with a few revisions and the alteration of the title. Best wishes.

Associate Editor Comments to Author (Dr Sean Rands):

Associate Editor

Comments to the Author:

Firstly, please accept my apologies for the extreme delay in getting a decision to you - I have been somewhat compromised by other duties associated with the ongoing pandemic, and the delay lies with me.

This is a nice revision, and both I and the reviewer who reread the manuscript agree that this is publishable. I'm suggesting this should be a minor revision as there are a few small cosmetic changes that need to be made - I have detailed how to change them, so these should not take you long at all.

TITLE. I think the 'dark eggs matter' part of the title may inadvertently cause offence to some readers, so please can you remove this (so the title should just be "Common cuckoo females remove more conspicuous eggs during parasitism")

42. change 'former parasitic cuckoos' to 'cuckoos who have already visited the nest'

48: preventive -> preventative

65: "birds; therefore, passing the high reproductive cost on to their hosts" -> "birds, passing the high reproductive cost of parental care on to their hosts"

83: supplement *nutrient* consumption

87: "to prevent" -> "as preventing"

113: less -> fewer

114: "they are not easy to find by later parasites" -> "they are not easy for later parasites to find"

294: "are easy to be" -> "could easily be"

297: former -> initial

SUPPLEMENTARY MATERIAL: I can't manage to extract the videos from this format. Could you either load the videos and data as uncompressed single files, or else zip them together in a different file type (.zip is probably more universal across different operating systems)

Reviewer comments to Author:

Reviewer: 1

Comments to the Author(s)

The authors have addressed most of the comments from the previous review and the manuscript is improved. I think it makes a good contribution to our understanding of cuckoo behaviour when removing eggs at host nests.

===PREPARING YOUR MANUSCRIPT===

===PREPARING YOUR REVISION IN SCHOLARONE===

-- If you have uploaded ESM files, please ensure you follow the guidance at <https://royalsociety.org/journals/authors/author-guidelines/#supplementary-material> to include a suitable title and informative caption. An example of appropriate titling and captioning may be found at https://figshare.com/articles/Table_S2_from_Is_there_a_trade-off_between_peak_performance_and_performance_breadth_across_temperatures_for_aerobic_scops_in_teleost_fishes_/3843624.

Author's Response to Decision Letter for (RSOS-201264.R0)

See Appendix B.

Decision letter (RSOS-201264.R1)

Dear Dr Wang,

It is a pleasure to accept your manuscript entitled "Common cuckoo females remove more conspicuous eggs during parasitism" in its current form for publication in Royal Society Open Science.

Best regards,

on behalf of Dr Sean Rands (Associate Editor) and Kevin Padian (Subject Editor)
openscience@royalsociety.org

Appendix A

Response to comments (RSOS-201264)

**Dear Dr Sean Rands and Kevin Padian,
Associate Editor and Subject Editor
Royal Society Open Science**

Thank you very much for kindly giving us the chance to revise and resubmit our manuscript (RSOS-201264).

We have read the comments carefully and revised the paper as suggested by you and the referee. This has resulted in a more balanced and focused paper, and we hope that you would agree with this assessment.

Please find explained in blue bold font below how we have addressed these points.

Yours sincerely,

on behalf of the co-authors,

Longwu Wang

Dear Dr Wang

On behalf of the Editors, we are pleased to inform you that your Manuscript RSOS-201264 "Dark eggs matter: common cuckoo females remove more conspicuous eggs during parasitism" has been accepted for publication in Royal Society Open Science subject to minor revision in accordance with the referees' reports. Please find the referees' comments along with any feedback from the Editors below my signature.

Please submit your revised manuscript and required files (see below) no later than 7 days from today's (ie 23-Nov-2020) date. Note: the ScholarOne system will 'lock' if submission of the revision is attempted 7 or more days after the deadline. If you do not think you will be able to meet this deadline please contact the editorial office

immediately.

on behalf of Dr Sean Rands (Associate Editor) and Kevin Padian (Subject Editor)
openscience@royalsociety.org

Editor comments:

Thanks for your resubmission; it will be acceptable with a few revisions and the alteration of the title. Best wishes.

Reply:

Thank you. We will revised the MS in accordance with the comments from editors and the referees.

Associate Editor Comments to Author (Dr Sean Rands):

Associate Editor

Comments to the Author:

Firstly, please accept my apologies for the extreme delay in getting a decision to you - I have been somewhat compromised by other duties associated with the ongoing pandemic, and the delay lies with me.

This is a nice revision, and both I and the reviewer who reread the manuscript agree that this is publishable. I'm suggesting this should be a minor revision as there are a few small cosmetic changes that need to be made - I have detailed how to change them, so these should not take you long at all.

Reply:

Thank you for your effort and help us improving the MS.

TITLE. I think the 'dark eggs matter' part of the title may inadvertently cause offence to some readers, so please can you remove this (so the title should just be "Common cuckoo females remove more conspicuous eggs during parasitism")

Reply:

Thank you. Removed, pleased see the title.

42. change 'former parasitic cuckoos' to 'cuckoos who have already visited the nest'

Reply:

Thank you. Changed. Please see line 42.

48: preventive -> preventative

Reply:

Thank you. Done.

65: "birds; therefore, passing the high reproductive cost on to their hosts" -> "birds, passing the high reproductive cost of parental care on to their hosts"

Reply:

Thank you. Reworded, please see line 65.

83: supplement *nutrient* consumption

Reply:

Thank you. Revised.

87: "to prevent" -> "as preventing"

Reply:

Thank you. Changed, please see line 88.

113: less -> fewer

Reply:

Thank you. Done.

114: "they are not easy to find by later parasites" -> "they are not easy for later parasites to find"

Reply:

Thank you. Done, please see lines 114-115.

294: "are easy to be" -> "could easily be"

Reply:

Done, please see line 294.

297: former -> initial

Reply:

Done.

SUPPLEMENTARY MATERIAL: I can't manage to extract the videos from this format. Could you either load the videos and data as uncompressed single files, or else zip them together in a different file type (.zip is probably more universal across different operating systems)

Reply:

We will load the videos and data as Zip file type.

Reviewer comments to Author:

Reviewer: 1

Comments to the Author(s)

The authors have addressed most of the comments from the previous review and the manuscript is improved. I think it makes a good contribution to our understanding of cuckoo behaviour when removing eggs at host nests.

Reply:

Thank you for your effort.

Appendix B

Response to comments (RSOS-201264)

**Dear Dr Sean Rands and Kevin Padian,
Associate Editor and Subject Editor
Royal Society Open Science**

Thank you very much for kindly giving us the chance to revise and resubmit our manuscript (RSOS-201264).

We have read the comments carefully and revised the paper as suggested by you and the referee. This has resulted in a more balanced and focused paper, and we hope that you would agree with this assessment.

Please find explained in blue bold font below how we have addressed these points.

Yours sincerely,

on behalf of the co-authors,

Longwu Wang

Dear Dr Wang

On behalf of the Editors, we are pleased to inform you that your Manuscript RSOS-201264 "Dark eggs matter: common cuckoo females remove more conspicuous eggs during parasitism" has been accepted for publication in Royal Society Open Science subject to minor revision in accordance with the referees' reports. Please find the referees' comments along with any feedback from the Editors below my signature.

Please submit your revised manuscript and required files (see below) no later than 7 days from today's (ie 23-Nov-2020) date. Note: the ScholarOne system will 'lock' if submission of the revision is attempted 7 or more days after the deadline. If you do not think you will be able to meet this deadline please contact the editorial office

immediately.

on behalf of Dr Sean Rands (Associate Editor) and Kevin Padian (Subject Editor)
openscience@royalsociety.org

Editor comments:

Thanks for your resubmission; it will be acceptable with a few revisions and the alteration of the title. Best wishes.

Reply:

Thank you. We will revised the MS in accordance with the comments from editors and the referees.

Associate Editor Comments to Author (Dr Sean Rands):

Associate Editor

Comments to the Author:

Firstly, please accept my apologies for the extreme delay in getting a decision to you - I have been somewhat compromised by other duties associated with the ongoing pandemic, and the delay lies with me.

This is a nice revision, and both I and the reviewer who reread the manuscript agree that this is publishable. I'm suggesting this should be a minor revision as there are a few small cosmetic changes that need to be made - I have detailed how to change them, so these should not take you long at all.

Reply:

Thank you for your effort and help us improving the MS.

TITLE. I think the 'dark eggs matter' part of the title may inadvertently cause offence to some readers, so please can you remove this (so the title should just be "Common cuckoo females remove more conspicuous eggs during parasitism")

Reply:

Thank you. Removed, pleased see the title.

42. change 'former parasitic cuckoos' to 'cuckoos who have already visited the nest'

Reply:

Thank you. Changed. Please see line 42.

48: preventive -> preventative

Reply:

Thank you. Done.

65: "birds; therefore, passing the high reproductive cost on to their hosts" -> "birds, passing the high reproductive cost of parental care on to their hosts"

Reply:

Thank you. Reworded, please see line 65.

83: supplement *nutrient* consumption

Reply:

Thank you. Revised.

87: "to prevent" -> "as preventing"

Reply:

Thank you. Changed, please see line 88.

113: less -> fewer

Reply:

Thank you. Done.

114: "they are not easy to find by later parasites" -> "they are not easy for later parasites to find"

Reply:

Thank you. Done, please see lines 114-115.

294: "are easy to be" -> "could easily be"

Reply:

Done, please see line 294.

297: former -> initial

Reply:

Done.

SUPPLEMENTARY MATERIAL: I can't manage to extract the videos from this format. Could you either load the videos and data as uncompressed single files, or else zip them together in a different file type (.zip is probably more universal across different operating systems)

Reply:

We will load the videos and data as Zip file type.

Reviewer comments to Author:

Reviewer: 1

Comments to the Author(s)

The authors have addressed most of the comments from the previous review and the manuscript is improved. I think it makes a good contribution to our understanding of cuckoo behaviour when removing eggs at host nests.

Reply:

Thank you for your effort.